# Luteolin-3′-*O*-Phosphate Inhibits Lipopolysaccharide-Induced Inflammatory Responses by Regulating NF-κB/MAPK Cascade Signaling in RAW 264.7 Cells

**DOI:** 10.3390/molecules26237393

**Published:** 2021-12-06

**Authors:** Jung-Hwan Kim, Tae-Jin Park, Jin-Soo Park, Min-Seon Kim, Won-Jae Chi, Seung-Young Kim

**Affiliations:** 1Department of Pharmaceutical Engineering & Biotechnology, Sunmoon University, Asan 31460, Korea; k5991313@naver.com (J.-H.K.); bark.taejin@gmail.com (T.-J.P.); 2Natural Product Informatics Research Center, Korea Institute of Science and Technology (KIST), Gangneung 25451, Korea; jinsoopark@kist.re.kr (J.-S.P.); nari7040@gmail.com (M.-S.K.); 3Genetic Resources Assessment Division, National Institute of Biological Resources, Incheon 22689, Korea; wjchi76@korea.kr

**Keywords:** luteolin-3′-*O*-phosphate, anti-inflammatory, MAPK, NF-κB, biorenovation

## Abstract

Luteolin (LT), present in most plants, has potent anti-inflammatory properties both in vitro and in vivo. Furthermore, some of its derivatives, such as luteolin-7-*O*-glucoside, also exhibit anti-inflammatory activity. However, the molecular mechanisms underlying luteolin-3′-*O*-phosphate (LTP)-mediated immune regulation are not fully understood. In this paper, we compared the anti-inflammatory properties of LT and LTP and analyzed their molecular mechanisms of action; we obtained LTP via the biorenovation of LT. We investigated the anti-inflammatory activities of LT and LTP in macrophage RAW 264.7 cells. We confirmed from previously reported literature that LT inhibits the production of nitric oxide and prostaglandin E_2_, as well as the expression of inducible NO synthetase and cyclooxygenase-2. In addition, expressions of inflammatory genes and mediators, such as tumor necrosis factor-α, interleukin-6, and interleukin-1β, were suppressed. LTP showed anti-inflammatory activity similar to LT, but better anti-inflammatory activity in all the experiments, while also inhibiting mitogen-activated protein kinase and nuclear factor-kappa B more effectively than LT. At a concentration of 10 μM, LTP showed differences of 2.1 to 44.5% in the activity compared to LT; it also showed higher anti-inflammatory activity. Our findings suggest that LTP has stronger anti-inflammatory activity than LT.

## 1. Introduction

Inflammation is a unique limiting process for the identification and destruction of invading pathogens; it is achieved through biological reactions that occur in the human body, and it restores normal tissue structure and function [1,2,3,4]. Normal inflammation is caused by an immune response to adverse stimuli or the upregulation of anti-inflammatory cytokines [5]. However, chronic inflammation is associated with an increased risk of tissue damage and the production of inflammatory cytokines [6]. The stimulation of macrophages with lipopolysaccharide (LPS) induces the production of nitric oxide (NO) and inflammatory cytokines, such as tumor necrosis factor (TNF-α), interleukin-6 (IL-6), and interleukin-1β (IL-1β) [7]. NO regulates the pathological and physiological conditions for reactive radicals produced by inducible nitric oxide synthase (iNOS) [8]. However, NO overexpression induces adverse reactions, such as acute or chronic inflammatory diseases and tissue damage [9]. Prostaglandin (PG) is produced in most cells in the body and synthesized in arachidonic acid through the action of cyclooxygenase (COX) enzymes as a stimulus to cells. Among them, PGE_2_ is the most prostanoid in the body and exhibits inflammatory or, in some cases, anti-inflammatory effects [10]. IL-6 and TNF-α can both induce the progression of inflammation via iNOS and COX-2 [11]. In addition, nuclear factor-kappa B (NF-κB) regulates the expression of pro-inflammatory cytokines when activated by inflammatory stimulation by LPS, a stimulant [12]. Mitogen activated protein kinase (MAPK) is known to include extracellular signal-regulated kinase (ERK), c-jun NH_2_-terminal kinase (JNK), and p38. MAPKs are involved in inflammatory and immune responses that are activated by stimuli such as LPS. Previous studies have been reported that MAPK plays an important role in the activation of NF-κB [13,14,15]. MAPK and NF-κB are important targets of anti-inflammatory reaction [16,17]. Therefore, discovering novel materials (or a novel material) to inhibit the activity of MAPK and NF-κB was undertaken in this study.

Active research and development are currently being carried out to increase the bioactivity of existing natural products and organic compounds using biorenovation technology [18,19]. Among these studies, attempts are being made to improve the biological activities of organic compounds using biorenovation, in which organic compounds are structurally transformed by inducing the enzyme-catalyzed reactions by microorganisms [20,21]. Luteolin (LT) is a natural flavonoid found in many plants; it is known to have powerful antioxidant and anti-inflammatory effects [22,23,24]. Plants with high LT contents have long been used in traditional medicine practices in Brazil, China, and Iran to treat inflammation-related diseases [25]. Hence, natural products are important for the development of new drugs; clinical trials are currently underway for more than 100 drugs derived from natural products [26]. Among flavonoids, LT, in particular, has been actively studied for its anti-inflammatory activity. It is also extracted from various plants for traditional purposes. 

Biorenovation methods have been used to study natural products that exhibit better anti-inflammatory activity than conventional substances, such as increasing cell viability or inhibiting NO production [27,28]. Among these products, it was hypothesized that LT, which is found in many natural products, could be converted using biorenovation to increase the inhibitory effect compared to its existing anti-inflammatory activity. It was confirmed that the biorenovation of LT produces luteolin-3′-*O*-phosphate (LTP).

The pharmacological mechanisms of LT and luteolin-7-*O*-glucoside have been revealed previously [29]. However, the biological activity and pharmacological mechanisms of LTP and LT have not yet been compared. Hence, we compared the differences in the anti-inflammatory activities of LT and LTP using RAW 264.7 cells and investigated the underlying molecular mechanisms of LTP activity.

## 2. Results

### 2.1. Analysis and Identification of LT Biorenovation Conversion Products

The LT biorenovation product separated through the ethyl acetate (EA) fraction was divided into an EA layer and a water layer, and HPLC analysis was performed. As a result, the peaks of the standard LT compound and LTP were confirmed in the water layer (Figure 1a). In the EA layer, the standard LT compound and three significant peaks were detected, but they did not show effective NO inhibitory activity (data not shown). We purified only LTP from the water layer, and the analysis of the LTP using electrospray ionization mass spectrometry (ESIMS) revealed a peak at m/z 286 (equivalent to LT) (Figure 1b). These results suggested that LTP is the molecular formula C_15_H_11_O_9_P as determined using high-resolution ESI/MS (HRESI/MS) and that this compound is in the form of phosphoric acid bound to LT. To our knowledge, this structural LTP has not been previously reported (Figure 1c).

### 2.2. Determination of Position of Phosphoryation with NMR Results

The phosphorylated product was elucidated by NMR spectroscopy data. The ^1^H-and 2D NMR experiments were done as previously reported [30]. The ^13^C data were obtained from the ^1^H, HSQC, and HMBC data. In the comparison to the ^1^H NMR spectrum of luteolin as the starting material, the spectrum of the product is characterized by a downfield shift of H-2′, H-5′, and H-6′ corresponding to the C-ring of flavonoid. In addition, the chemical shift of C-3′ in the product was significantly upfield-shifted to 141.47 ppm compared to that of luteolin (145.80 ppm), indicating that the phosphorylation occurred at 3′-OH of luteolin. Furthermore, more deshielded adjacent carbons including C-2′ and C-4′ convinced this phosphorylation. 

Luteolin-3′-*O*-phosphate (LTP): ^1^H-NMR (500 MHz, DMSO) δ 7.71 (1H, s, H-2′), 7.69 (1H, d, H-6′), 6.97 (1H, d, H-5′), 6.73 (1H, s, H-3), 6.47 (1H, d, H-8), 6.2 (1H, d, H-6). ^13^C-NMR (125 MHz, DMSO) δ 182.29 (C-4), 164.89 (C-7), 163.81 (C-2), 162.1 (C-5), 157.8 (C-9), 153.93 (C-4′), 141.47 (C-3′), 123.83 (C-6′), 121.7 (C-1′), 120.28 (C-2′), 118.67 (C-5′), 104.3 (C-10), 103.68 (C-3), 99.49 (C-6), 94.5 (C-8) (Appendix A).

### 2.3. Cytotoxic Effects of Compounds on RAW 264.7 Cells

The MTT assays showed that either LT or LTP products exhibited greater than 89% cell viability, and none were cytotoxic at concentrations of 1, 5, or 10 μM (Figure 2).

### 2.4. Production of NO and PGE_2_

NO and PGE_2_ are sub-mechanisms of iNOS and COX-2 and are known to be end products of inflammation. As shown in Figure 3a, both LT and LTP treatments reduced NO production. However, LTP inhibited the NO production more than LT. Regarding the measurement of the PGE_2_ levels, LT showed inhibition rates of 35.8, 49.8, and 68.1%, whereas LTP showed inhibition rates of 56.4, 60.8, and 70.2% at 1, 5, and 10 μM, respectively (Figure 3b). These results indicated that LTP showed a higher PGE_2_ inhibition rate than LT at low concentrations.

### 2.5. Comparison of iNOS and COX-2 Expression Inhibition

Both LT and LTP decreased the iNOS expression and COX-2 expression in a concentration-dependent manner, but LTP showed a higher reduction rate than LT at the same experimental concentration (Figure 4a,b). Comparing the inhibition rates of LT and LTP revealed that LTP inhibited iNOS and COX-2 expression more than LT in a dose-dependent manner. These results show that the reductions in the iNOS and COX-2 expression led to the observed NO and PGE_2_ production reductions.

### 2.6. Effects of LT and LTP on LPS-Induced IL-1β, IL-6, and TNF-α Expression

At concentrations of 1, 5, and 10 μM, LT and LTP decreased the expressions of TNF-α, IL-1β, and IL-6 in a concentration-dependent manner, as shown in Figure 5. In particular, 10 μM of LT and LTP inhibited the IL-1β expression by 52.7 and 55.2% (Figure 5b) and IL-6 expression by 46.3 and 51.4%, respectively (Figure 5c). Collectively, these results indicate that LTP exhibited an anti-inflammatory effect by suppressing the expressions of the proinflammatory cytokines TNF-α, IL-1β, and IL-6 more compared to LT.

### 2.7. Effects of LT and LTP on the Activation of NF-κB and MAPK

NF-kB is known to regulate the expression of proinflammatory cytokines. Moreover, MAPK is involved in inflammation and plays an important role in the activation of NF-kB. As a result of understanding the MAPK regulation by analyzing p38, ERK, and JNK, as shown in Figure 6, LT slightly inhibited p38, whereas LTP decreased MAPK in a concentration-dependent manner. In particular, it was shown that p38 and ERK were significantly reduced by LTP. IκB-α, which rapidly changes NF-κB activation, is phosphorylated and degrades IκB-α to phospho IκB-α. Phospho IκB-α is known to decrease the inflammatory response with increased expression. LTP more potently inhibited NF-κB and increased the expression of IκB-α more effectively than LT in a concentration-dependent manner. These results show that the reduction in the inflammatory factors and mediators increased by LPS in RAW 264.7 cells is regulated by MAPK and NF-κB inhibition is regulated by LTP.

## 3. Discussion

Here, we attempted to develop a new anti-inflammatory agent through biorenovation using LT as a substrate. We used MS and NMR to identify this compound as LTP. We then investigated the anti-inflammatory activity of LTP. In RAW 264.7 cells stimulated by LPS, both LT and LTP slightly decreased the levels of pro-inflammatory factors and kinases but, at the same time, showed different mechanisms of action. In conclusion, LTP inhibited the activation of MAPK-dependent inflammation more strongly than LT.

Inflammation refers to how tissue responds when it is damaged. It can be caused by an increase in inflammatory cytokines, free radical production, or LPS secretion. The overproduction of NO by iNOS and PGE_2_ by COX-2 is known as a common reaction to inflammation [31]. In the present study, we found that both LT and LTP decreased NO and PGE_2_ production in a dose-dependent manner. As a result of confirming iNOS and COX-2, which are known as upstream mechanisms of NO and PGE_2_, through Western blot, it was confirmed that they decreased in a concentration-dependent manner. We found that, at the experimental concentration, LTP inhibited the production of NO and PGE_2_ and inhibited the expression of iNOS and COX-2 more than LT. TNF-α, IL-6, and IL-1β, known as proinflammatory cytokines, play an important role in inducing tissue damage and mediating several inflammatory diseases [32]. LT was found to decrease the secretion of TNF-α, IL-6, and IL-1β in a previous study [33]. As shown in Figure 4, LTP also decreased the secretion of TNF-α and IL-6. As a result of confirming the production rates of TNF-α, IL-6, and IL-1β, LTP showed higher inhibition than LT in all the experiments. These results suggest that LTP exhibits stronger anti-inflammatory activity than LT. The transcriptional regulator NF-κB plays an important role in the inflammatory response because it has the ability to induce the transcription of gene sequences associated with inflammation, so it can specifically induce the regulation of pro-inflammatory molecules [34]. NF-κB forms the inhibitor IκB under normal physiological conditions. When IκB is phosphorylated, NF-κB translocates to the nucleus to activate the target gene [35]. In contrast to IκB-α degradation, which leads to a sharp change in NF-κB activation, IκB-b degradation is known to be associated with prolonged NF-κB activation [36,37]. Furthermore, the activation of NF-κB not only affects iNOS and COX-2 but also affects several genes involved in inflammatory action [38]. LT is known to significantly inhibit NO and PGE_2_ production due to NF-κB and active protein-1 inactivation [29,33].

Here, we identified MAPK and NF-κB via Western blotting to compare the signaling pathways for the anti-inflammatory effects of LT and LTP. We found that LTP decreased NF-κB phosphorylation and increased IκB-α expression more effectively than LT at 10 μM, suggesting that LTP inhibits IκB-α and inactivates NF-κB, thereby suppressing the expressions of iNOS and COX-2. Among the MAPKs, p38 and JNK are stress-activated protein kinases that play essential roles in apoptosis and inflammation [39]. This signaling pathway activates AP-1 and NF-κB [40]. We examined MAPK to confirm further the mechanism of NF-κB inactivation for LT and LTP, revealing that LTP exhibited higher inhibition rates than LT for p38, JNK, and ERK. In particular, LTP effectively inhibited p38, unlike LT, which had a minimal inhibitory effect. Overall, LTP downregulated the p38, JNK, and ERK signaling pathways between MAPKs more than LT and inhibited NF-κB.

We confirmed that the reduction in the production of iNOS, COX-2, and pro-inflammatory cytokines induced by LPS was caused by the regulation of the MAPK and NF-κB signaling pathways through the experimental results. Both LT and LTP showed anti-inflammatory activity because they regulate the same signaling pathways, but LTP showed more effective inhibitory activity than LT on p38, ERK, NF-κB, and IκB. This result suggests that the higher anti-inflammatory effect of LTP compared to that of LT was because of its more superior inhibitory activity on the MAPK and NF-κB pathways compared to LT. LPS-induced NF-κB and AP-1 activation have been reported to be regulated by a series of events that lead to the activation of MAPK and Akt [41,42,43]. Thus, we suggest that further studies comparing Akt phosphorylation inhibitory activity are needed to understand the signaling pathways involved in the anti-inflammatory activities of LT and LTP. Our study confirmed that a material synthesized through biorenovation exhibited higher anti-inflammatory activity than that of its substrate. Our findings also suggest that LTP, a compound synthesized through biorenovation, can be used to develop effective anti-inflammatory drugs.

## 4. Materials and Methods

### 4.1. Reagents and Strains Used in Biorenovation

LT was obtained from Sigma-Aldrich (St. Louis, MO, USA). Nutrient medium components were obtained from Difco (Baltimore, MD, USA). Bacillus sp. JD3-7 strain (KCTC92346P) was obtained from the Korean Collection for Type Cultures (Seoul, Korea).

### 4.2. Biorenovation of LT

For the seed culture, Bacillus sp. JD3-7 colonies were placed in a culture tube containing 4 mL of nutrient broth (peptone [ThermoFisher, Waltham, MA, USA] and beef extract [ThermoFisher]) and cultured for 16 h. Then, 100 mL of nutrient medium was placed in a 500 mL flask, adding 0.2% of the total volume to Bacillus sp. JD3-7 cells were added. The flask was then incubated for 16 h, shaking at 150 rpm in an incubator at 30 °C. The culture medium was then centrifuged at 3577× *g* for 15 min; only the bacterial pellet was recovered. The obtained pellet was washed twice with PG buffer (50 mM phosphate buffer, 2% glycerol, pH 7.2). The washed bacterial pellet, PG buffer, and 50 mg of LT were then placed in a 500 mL flask and incubated for 72 h, shaking at 150 rpm in an incubator at 30 °C. After incubation, centrifugation was performed at 4416× *g* for 10 min using a centrifuge, and the divided supernatant was freeze-dried and powdered. For more accurate separation and identification, ethyl acetate (EA, DAEJUNG, Busan, Korea) fractionation was performed. The sample was dissolved in purified water, and twice the amount of EA was added to separate the layers. The water layer and the EA layer were recovered separately and repeated 3 times. The separated layers were each concentrated and sampled.

### 4.3. HPLC Analysis of the LT Biorenovation Product

HPLC analysis was performed using a Shimadzu SpectroMonitor 3200 equipped with a Shim-pack GIS 0.5 mm ODS C18 column (250 × 4.6 mm id), with the mobile phase consisting of solvent A (0.1% *v*/*v* trifluoroacetic acid in water) and B (acetonitrile). A gradient method was used in which the flow rate of solvent B (1 mL/min) was increased from 10/90 to 50/50 over 30 min. The column temperature was maintained at 40 °C. The wavelength was measured at 254 nm.

### 4.4. Liquid Chromatography–Mass Spectrometry (LC–MS) and Nuclear Magnetic Resonance (NMR) of the LT Biorenovation Product

LC/MS analysis of LTP was confirmed in positive ion mode using Agilent 1260 Infinity II (Agilent, Santa Clara, CA, USA). Luna^®^ C18(2) (2 mm × 100 mm, 3 μm particle size) columns were used for the analysis. 5 μL of LT and LTP were injected into the C18 column and eluted at a flow rate of 300 μL/min. The mobile phase was composed of solvent A (0.1% *v*/*v* formic acid in water) and B (acetonitrile) and increased from 10/90 to 5/95 over 12 min. The column temperature was maintained at 40 °C. The wavelength was measured at 254 nm. The spectrum used for NMR analysis was prepared from a VNMRS 500 NMR spectrometer (Agilent Technology, Santa Clara, CA, USA), and residual solvent peaks (DMSO-d_6_ = δ_H_ 2.50) of deuterated NMR solvent (Sigma-Aldrich, St. Louis, MO, USA) were used as a reference peak.

### 4.5. Cell Culture, Treatment, and Cell Viability Assay

The RAW 264.7 cell line was purchased from the Korean Cell Line Bank (Seoul, Korea) and cultured in Dulbecco’s modified Eagle’s medium (Welgene, Gyeongsan, Korea); 10% fetal bovine serum (Welgene, Gyeongsan, Korea) and 1% penicillin were added. Cells in 24-well plates (7 × 10^4^ cells per well) were incubated for 24 h with the indicated concentrations of LT and LTP (1, 5, and 10 μM) with or without LPS (1 μg/mL) in a humid atmosphere containing 5% CO_2_ at 37 °C. Cell viability was measured using a 3(4,5-dimethyl-2-thiazolyl)-2,5-diphenyl tetrazolium bromide (MTT) assay [44]. Cultured cells were incubated with the MTT reagent (1 mg/mL) at 37 °C for 4 h. After removing the media, the formazan crystals were dissolved in dimethyl sulfoxide and placed onto a 96-well plate at 100 μL cells per well. Absorbance was measured via a microplate reader with the wavelength value set to 570 nm (Spectrophotometer, ThermoFisher, Waltham, MA, USA).

### 4.6. Nitrite and Prostaglandin E_2_ (PGE_2_) Level Determination

RAW 264.7 cells were seeded onto 24-well plates (7.0 × 10^4^ cells per well) and incubated for 24 h. Then, with or without 1 μg/mL of LPS, various concentrations of either LT or LTP (1, 5, 10 μM) were added, and plates were incubated for 24 h for NO production. Griess reaction was used to measure the amount of NO accumulated in the medium. The PGE_2_ concentration in the cell culture supernatant was measured using an enzyme-linked immunosorbent assay (ELISA) kit (R&D Systems Inc., Minneapolis, MN, USA).

### 4.7. Measurement of pro-Inflammatory Cytokine Production (TNF-α, IL-1β, and IL-6)

RAW 264.7 cells were seeded onto 24-well plates (7.0 × 10^4^ cells per well) and incubated for 24 h. Then, with or without 1 μg/mL of LPS, various concentrations of either LT or LTP (1, 5, 10 μM) were added, and plates were incubated for 24 h. Supernatants were used for pro-inflammatory cytokine assays using a mouse ELISA kit (R&D Systems Inc., Minneapolis, MN, USA).

### 4.8. Western Blot Analyses

Whole-cell proteins from each sample were extracted using radioimmunoprecipitation assay buffer (BioRad, Hercules, CA, USA) and assayed with a Bradford assay kit (Pierce BCA Protein Assay Kit, Thermo, Waltham, MA, USA). After analysis, 20 μg of protein was resolved in each sample by electrophoresis using a 10% sodium dodecyl sulfate-polyacrylamide gel. The digested proteins were transferred to polyvinylidene difluoride (PVDF) membranes (BioRad, Hercules, CA, USA) at 130 V for 2 h. Following the transfer, the membranes were incubated in 5% skim milk (BD, Franklin Lakes, NJ, USA) to block non-specific binding during subsequent immunostaining. Then, the membranes were incubated overnight at 4 °C with antibodies against iNOS, COX-2, p-JNK, p-NF-κB, p-IκB-α, p-ERK, and p-p38. Membranes were washed 3 times with 10 min intervals using 10x Tris-buffered saline with Tween (TBST) and then incubated with the corresponding horseradish peroxidase (HRP)-conjugated secondary antibody for 1 h at room temperature. The membranes were then washed three times with 10× TBST at 10 min intervals, exposed to an enhanced chemiluminescence kit (BioRad, Hercules, CA, USA), and visualized on an image reader (LAS-4000, Fujifilm, Japan). After detection, the cells were incubated with antibodies against JNK, NF-κB, IκB-α, ERK, p38, and β-actin. Immunostaining was repeated as described above. Band densities were determined after measurement using the ImageJ analysis program.

### 4.9. Statistical Analysis

All results were expressed as mean ± standard deviation (SD) of the results of at least three independent experiments. The data obtained through the experiment were expressed by evaluating the statistical significance of the difference using Student’s *t*-test.

## Figures and Tables

**Figure 1 molecules-26-07393-f001:**
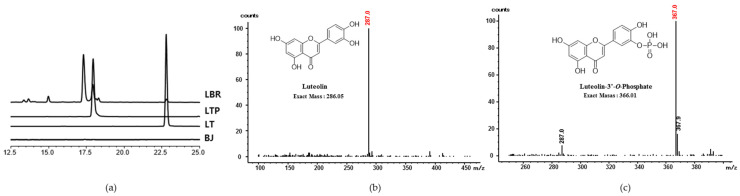
(**a**) HPLC analysis of biorenovation negative control (BJ), luteolin biorenovation supernatant (LBR), luteolin (LT), and luteolin-3′-*O*-phosphate (LTP). (**b**,**c**) Mass spectrometry of LT and LTP.

**Figure 2 molecules-26-07393-f002:**
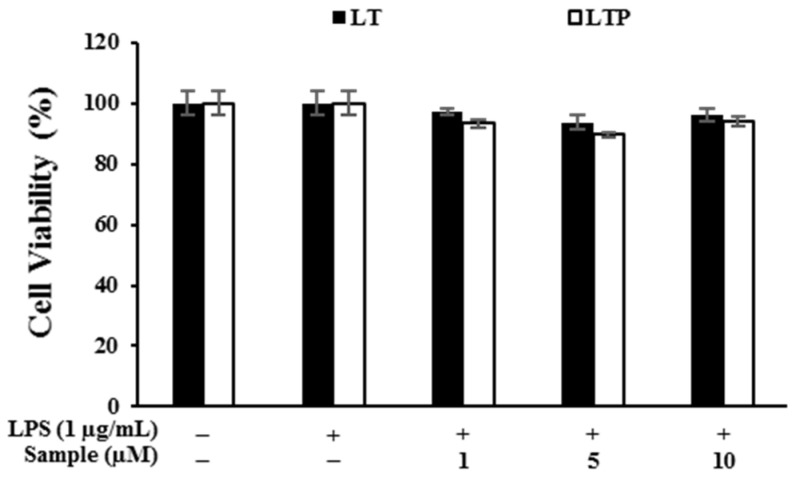
Viability of cells stimulated with LPS (1 μg/mL) for 24 h in the presence of LT or LTP. Data are expressed as mean ± standard deviation (SD) from three independent experiments (compared to the LPS-non-treated group).

**Figure 3 molecules-26-07393-f003:**
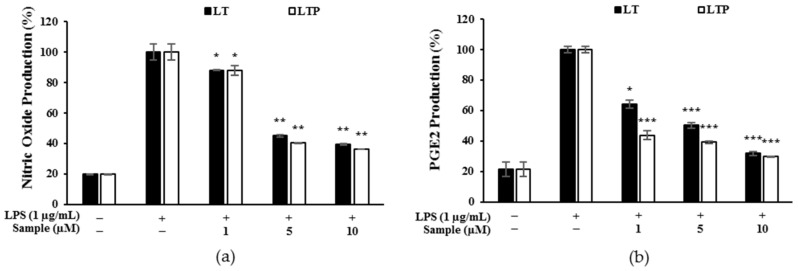
Effects of LT and LTP (at 1, 5, and 10 μM) on (**a**) nitric oxide (NO) and (**b**) prostaglandin E2 (PGE_2_) production in lipopolysaccharide (LPS)-stimulated RAW264.7 cells. Data are expressed as mean ± SD from three independent experiments (* *p* < 0.05, ** *p* < 0.01, and *** *p* < 0.001, compared to the LPS-treated group).

**Figure 4 molecules-26-07393-f004:**
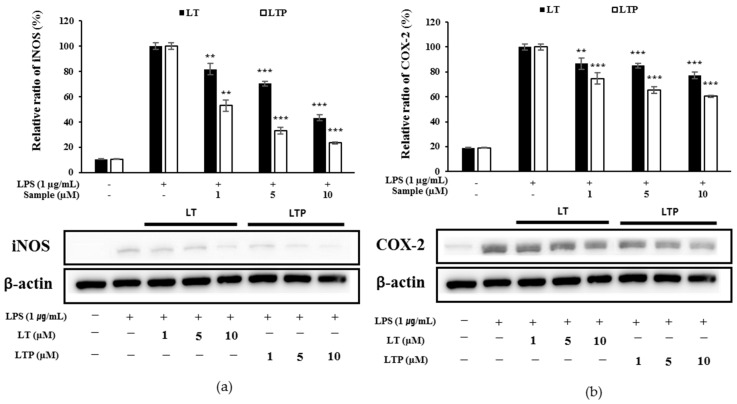
(**a**) Inducible NO synthase (iNOS) and (**b**) cyclooxygenase-2 (COX-2) expression upon LT and LTP treatment in LPS-stimulated RAW264.7 cells. Data are expressed as mean ± SD from three independent experiments (** *p* < 0.01, and *** *p* < 0.001, compared to the LPS-treated group).

**Figure 5 molecules-26-07393-f005:**
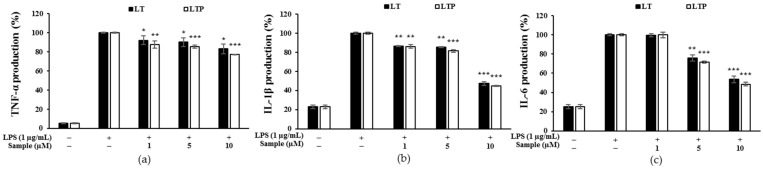
Comparison of the inhibitory effects of LT and LTP on LPS-induced (**a**) tumor necrosis factor-α (TNF-α), (**b**) interleukin-1β (IL-1β), and (**c**) interleukin-6 (IL-6) production in RAW 264.7 macrophages. Data are expressed as mean ± SD from three independent experiments (* *p* < 0.05, ** *p* < 0.01, and *** *p* < 0.001, compared to the LPS-treated group).

**Figure 6 molecules-26-07393-f006:**
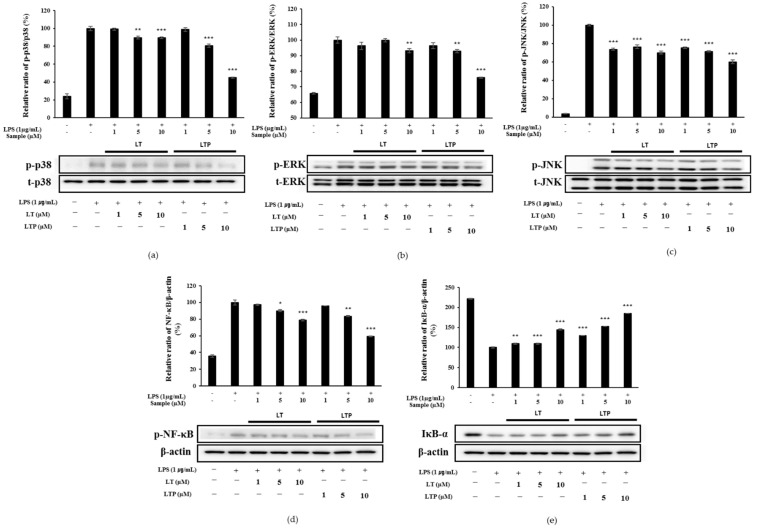
Effects of LT and LTP on the expression levels of mitogen-activated protein kinase (MAPK) and nuclear factor-kappa B (NF-κB) in LPS-stimulated RAW 264.7 cells for (**a**) p38, (**b**) extracellular signal-regulated kinase (ERK), (**c**) c-jun NH_2_-terminal kinase (JNK), (**d**) nuclear factor-kappa B (NF-κB), and (**e**) IκB-α. Data are expressed as mean ± SD from three independent experiments (* *p* < 0.05, ** *p* < 0.01, and *** *p* < 0.001, compared to the LPS-treated group).

## Data Availability

Not applicable.

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
