# Peer review of "Luteolin-3′-O-Phosphate Inhibits Lipopolysaccharide-Induced Inflammatory Responses by Regulating NF-κB/MAPK Cascade Signaling in RAW 264.7 Cells"

_molecules, 2021, doi:10.3390/molecules26237393_

Round 1

Reviewer 1 Report

The article of Jung-Hwan Kim et. al. entitled “Luteolin-4’-O-phosphate inhibits Lipopolysaccharide-induced inflammatory responses by regulating NF-κB/MAPK cascade signaling in RAW 264.7 cells” describes an informative study that aims to highlight that Luteolin-4’-O-phosphate has potential anti-inflammatory activity, however, some concerns should be addressed by the Authors.

  • The authors should provide their own justification and relevance of the study. Relevant articles in the field such as (PMID: 22203870; PMID: 21782879; PMID: 21513709; PMID: 33446171; PMID: 20562902; PMID: 29497773; PMID: 30119240). The above studies have been comprehensively explained Luteolin and their derivatives inhibits inflammatory responses by regulating NF-κB/MAPK or other cascade signaling pathways. Hence, there is no novelty of the study

Reviewer 2 Report

This is, on the whole, a good manuscript. However, I have major concerns about the assignment of the structure of the phosphorylated Luteolin (LT). Authors do NOT provide any information, or evidence to support assignment of the product of biorenovation as the 4' isomer. According to the C-13 data, C-4' of LT has a chemical shift of 149.7 but after phosphorylation, the chemical shift of the same carbon increases to 154 in LTP. However, phosphorylation should result in a decrease in the chemical shift of C-4'. How can authors explain this? what is the evidence that this is the 4' O-phosphate, since the authors state this compound is novel?

In addition, authors should pay attention to the English language when resubmitting their manuscript. Lines 79-80 does not make sense.

Finally, the authors should mention why they think LTP is more efficacious than LT and whether this has any physiological significance.  for instance, are there kinases that can affect this in the body?

Author Response

We tried to do our best to give an answer of reviewer’s opinions. In addition we are unanimously honored that International Journal of Molecular Science has given us a chance to evaluate our research in a world-wide journal. Thank you for your consideration and kindness.

This is, on the whole, a good manuscript. However, I have major concerns about the assignment of the structure of the phosphorylated Luteolin (LT). Authors do NOT provide any information, or evidence to support assignment of the product of biorenovation as the 4' isomer. According to the C-13 data, C-4' of LT has a chemical shift of 149.7 but after phosphorylation, the chemical shift of the same carbon increases to 154 in LTP. However, phosphorylation should result in a decrease in the chemical shift of C-4'. How can authors explain this? what is the evidence that this is the 4' O-phosphate, since the authors state this compound is novel?

- We deeply apologize for the error of structural assignment in the manuscript. During the writing of the manuscript, 3’-O-phosphate was incorrectly described as 4’-O-phosphate. In short, 3’-O-phosphate was correct. As your valuable comment, C-3’, which is bound to phosphate was upfield-shifted from 145.80 ppm to 141.47 ppm. In addition, adjacent carbons and protons were all downfield-shifted. This kind of NMR data shows unambiguous phosphorylation at C-3’ of luteolin. In addition, we have carefully revised the manuscript to correct the structural mistake. Thanks again for pointing out the important error.

In addition, authors should pay attention to the English language when resubmitting their manuscript. Lines 79-80 does not make sense.

- The LT biorenovation product separated through the Ethyl Acetate (EA) fraction was divided into an EA layer and a Water layer, and HPLC analysis was performed. As a result, the peaks of the standard LT compound and LTP were confirmed in the Water layer. In the lines 84, as reviewer’s pointed out, we completely fixed it.

Finally, the authors should mention why they think LTP is more efficacious than LT and whether this has any physiological significance.  for instance, are there kinases that can affect this in the body?

- Luteolin has been studied for a long time and has been used as a potent anti-inflammatory agent and its signaling pathway has already been elucidated. However, when comparing the MAPK and NF-κB of LT and LTP, unlike LT, LTP suppressed the expression of p38, ERK, JNK, and NF-κB and increased the expression of IkB-a. In particular, there was a clear difference in the NF-κB pathway. In addition, the absorption rate of LT was low due to weak water solubility, whereas the absorption rate of LTP was improved due to the binding of phosphate. I think this can increase the absorption rate in the body. In addition, when phosphate is bound, it has the advantage that the structure is not destroyed when it enters the cell. The following are the results of research conducted to increase the absorption rate by increasing the water solubility of flavonoids.

R 1)

Juanjuan Zhao, Jun Yang, Yan Xiea

Improvement strategies for the oral bioavailability of poorly water-soluble flavonoids: An overview.

Int J Pharm. (2019) 30; 570:118642.

DOI: 10.1016/j.ijpharm.2019.118642

R 2)

Nemoto H, Cai J, Asao N, Iwamoto S, Yamamoto Y.

Synthesis and biological properties of water-soluble p-boronophenylalanine derivatives. Relationship between water solubility, cytotoxicity, and cellular uptake.

J Med Chem. 1995;38(10):1673-1678. DOI: 10.1021/jm00010a012

R 3)

Zhang S, Li DD, Zeng F, et al.

Efficient biosynthesis, analysis, solubility and anti-bacterial activities of succinylglycosylated naringenin.

Nat Prod Res. 2018;5(12):1-5. https://doi.org/10.1080/14786419.2018.1431633

Reviewer 3 Report

This article is devoted to the comparative study anti-inflammatory activities of luteolin (LT) and luteolin-4-O-phosphate (LTP) using RAW 264.7 cells with inductor LPS.

The authors conducted a large number of experiments to prove LTP is a more promising anti-inflammatory compound than LT. However, in my opinion, no significant differences have been revealed in any model, so it is worth asking the question of how justified the production of LTP will be, even by bio-renovation, and will LTP have any advantages over LT in in vivo experiments? How to explain that different effects of compounds on PGE2 and NO level production are observed at concentration 1 and 5 mkM, while the effect on MAPK is only at a concentration of 10 μM for LTP?

The manuscript can not be published in present form and it needs to be reworked.

Lines 16-19: These two sentences have the same sense, rephrase please

Lines 22-24: It is not clear, what is the semantic difference between “behavior” and “activity”? Why are these two terms opposed? Rephrase please

Lines 18, 28…: the term ‘Biorenovation’ should always be spelled the same

Lines 49-51: You cannot inhibit the activity of MAPK and NF-κB, you can find substances that inhibit ones, are that what you mean? Rephrase please

Line 85 Formula of LTP must be written correctly with lower subscripts

Line 114 When compare both compounds, there is no significant difference in NO inhibition. The values of NO inhibition like for PGE2 is not enough.

Line 115 The decryption of some terms like PGE2, EA (line 79), IκB-α (line 152) which mentioned firstly in the text is lack.

Line 144 The figure caption is not correctly presented

Line 149 This section must be expanded. A clearer description of the results obtained, indicating the concentrations of compounds in which inhibition is observed is needed. There is a lack of general explanation for p38 and IκB-α either in the introduction or in this section.

Lines 165-167 It is a strange conclusion. The authors showed that both compounds reduce, to a greater or lesser extent, the levels of pro-inflammatory factors and kinases, but at the same time have a different mechanism of action. In this case, LTP is a stronger inhibitor of the activation of MAPK-dependent inflammation than LT.

Lines 172-175 This sentence should be moved to Results

Lines 181-182 This sentence should be moved to Results

The reference list is not arranged according to the rules. Also, there are no doi for articles.
